# Comparative Analysis of Bio-Vanillin Recovery from Bioconversion Media Using Pervaporation and Vacuum Distillation

**DOI:** 10.3390/membranes12080801

**Published:** 2022-08-19

**Authors:** Rita Valério, Carla Brazinha, João G. Crespo

**Affiliations:** 1LAQV-Requimte, Department of Chemistry, NOVA School of Science and Technology, NOVA University of Lisbon, 2829-516 Caparica, Portugal; 2UCIBIO—Applied Molecular Biosciences Unit, Department of Chemistry, School of Science and Technology, NOVA University Lisbon, 2819-516 Caparica, Portugal

**Keywords:** bio-vanillin purification, extract purification, pervaporation-fractionated condensation, vacuum distillation–pervaporation-fractionated condensation

## Abstract

The increasing demand for natural products has led to biotechnological vanillin production, which requires the recovery of vanillin (and vanillyl alcohol at trace concentrations, as in botanical vanillin) from the bioconversion broth, free from potential contaminants: the substrate and metabolites of bioconversion. This work discusses the recovery and fractionation of bio-vanillin, from a bioconversion broth, by pervaporation and by vacuum distillation, coupled with fractionated condensation. The objective was to recover vanillin free of potential contaminants, with maximised fluxes and selectivity for vanillin against water and minimised energy consumption per mass of vanillin recovered. In vacuum distillation fractionated condensation, adding several consecutive water pulses to the feed increased the percentage of recovered vanillin. In pervaporation-fractionated condensation and vacuum distillation-fractionated condensation processes, it was possible to recover vanillin and traces of vanillyl alcohol without the presence of potential contaminants. Vacuum distillation–experiments presented higher vanillin fluxes than pervaporation fractionated condensation experiments, 2.7 ± 0.1 g·m^−2^ h^−1^ and 1.19 ± 0.01 g·m^−2^ h^−1^, respectively. However, pervaporation fractionated condensation assures a selectivity of vanillin against water of 4.5 on the pervaporation step (acting as a preconcentration step) and vacuum distillation fractionated condensation requires a higher energy consumption per mass of vanillin recovered when compared with pervaporation– fractionated condensation, 2727 KWh kg_VAN_^−1^ at 85 °C and 1361 KWh kg_VAN_^−1^ at 75 °C, respectively.

## 1. Introduction

Vanillin is the world’s most popular flavour and is commonly used in food, beverages, perfume, cosmetics and pharmaceutical products. Most vanillin is chemically produced, mainly from guaiacol and lignin [1]. Nowadays, there is an increasing consumer interest and market demand for healthy, natural food products. However, natural vanillin extracted from the beans of vanilla flowers cannot keep up with this increasing demand in terms of quantity and affordable price, contributing with less than 1% to the overall vanillin market [2,3]. Bio-vanillin is a good alternative to the vanillin extracted from vanilla flowers, since it is considered natural, according to US and European food legislation (U.S Food and Drug Administration [4] and Regulation (EC) No 1334/2008 [5], respectively).

The global bio-vanillin market is increasing, due to bio-vanillin’s excellent flavouring properties. In 2018, this market was valued at USD 150 million, and it is expected to reach USD 400 million by the end of 2025 [6], leading bio-vanillin companies to increase their production capacity [7]. Bio-vanillin can be obtained from the biotransformation of *natural* substrates by fungi, bacteria, genetically modified microorganisms and plant cells [2]. Ferulic acid is the most widely studied precursor for bio-vanillin production [2,8], mainly using the bacteria *Amycolatopsis* sp. ATCC 39116 due to its ability to tolerate higher levels of vanillin [9].

One of the challenges of vanillin bio-production is the recovery/purification of vanillin from the complex bioconversion broth. Crystallisation is perhaps the most extensively employed technique for the recovery and purification of bio-products from bioconversion broths [10]; however, in the case of vanillin, it only becomes a competitive technique when the concentration of vanillin is higher than 10 g·L^−1^ (minimum concentration to allow its crystallisation at 20 °C) [11]. Adsorption is another extensively studied process for vanillin recovery [12,13,14,15]. However, adsorption exhibits some disadvantages: (1) some resins and, in particular, activated carbon is not very selective to vanillin and, therefore, other compounds present in the bioconversion broth are also adsorbed, especially the metabolites of vanillin bioconversion (vanillic acid, vanillyl alcohol and guaiacol), as well as the substrate, ferulic acid [13,16,17,18]; (2) traces of adsorbent contaminants are difficult to avoid, requiring strict quality control of the recovered product [19]; and (3) organic solvents are normally required for the desorption of vanillin from the adsorbent.

Other processes have been reported, such as liquid–liquid extraction [20], but these processes use (non-sustainable) organic solvents, such as isopropyl acetate, methyl ethyl ketone, cyclohexane and dichloromethane. Membrane-based solvent extraction has been also proposed but the solvents used are typically the same [21,22,23]. Alternatively, evaporative processes have been studied and offer the main advantage of not using organic solvents.

Pervaporation was also proposed, mainly using PEBA (polyether block amide) membranes [24,25,26] and POMS (polyoctylmethylsiloxane) membranes [19]. Camera et al. [24,26] studied pervaporation with a PEBA membrane, coupled with photocatalysis for the green synthesis of vanillin from ferulic acid, to recover vanillin from the reactor with the minimum possible concentration of by-products. The PEBA membrane had a high rejection of all compounds present in the photocatalytic reactor, except vanillin and 4-vinyl guaiacol [24,26]. Camera et al., 2019 compared pervaporation with dialysis for the recovery of vanillin from a photocatalytic reactor and the results showed that pervaporation is much more selective than dialysis. In fact, using pervaporation, only vanillin and 4-vinyl guaiacol permeated the membrane, whereas, when using dialysis, ferulic acid and caffeic acid were also detected in the permeate [24]. However, the vanillin flux of dialysis is much higher than in pervaporation [24]. Brazinha et al., 2011 reported the recovery of pure vanillin from a bioreactor broth containing ferulic acid (used as a precursor for biovanillin production), using pervaporation (POMS membrane). The vanillin was selectively condensed at mild temperatures [19]. Böddeker et al. [25] reported pervaporation using a PEBA membrane to purify vanillin from a model bioconversion broth.

Vacuum distillation has been also reported to purify vanillin from agro-industrial hydrolysates [27,28,29]. Rhodia (Solvay) [30] used vacuum distillation to purify vanillin from the bioconversion broth. The Rhodia patent includes a vanillin purification process with one or more distillation columns and purification in an evaporator followed by condensation.

Both pervaporation and vacuum distillation technologies, combined with fractionated condensation, will be evaluated and compared in terms of the objective of this work (recovery of pure vanillin from bioconversion media, maximised vanillin fluxes and selectivity of vanillin against water and minimised energy consumption per mass of vanillin recovered).

A pervaporation-fractionated condensation process has been shown to recover pure vanillin from a bioconversion broth, using a POMS polyoctylmethylsiloxane membrane from GKSS [19]. Pervap 4060 polydimethylsiloxane was used on the pervaporation-fractionated condensation process, as Pervap 4060 was reported to perform better on organophilic pervaporation processes than POMS membranes because the former exhibited a higher hydrophobicity expressed by a higher surface free energy (SFE) [31].

In vacuum distillation–condensation fractionation experiments, the addition of several consecutive water pulses to the feed during the process significantly increased the percentage of recovered vanillin, which has not yet been reported.

## 2. Materials and Methods

### 2.1. Materials

The selected membrane for the pervaporation experiments was the commercial hydrophobic (organophilic) pervaporation membrane PervapTM 4060, of which the main characteristics are summarised in Table 1. This membrane is appropriate for pervaporation once it is dense, without pores.

The feed used in this work was the vanillin bioconversion broth produced in two bioreactors, operating under the same conditions, obtained according to the biochemical pathway of ferulic acid conversion (in this case commercial ferulic acid) into vanillin by the strain *Amycolatopsis* sp. ATCC 39116 [32], which is the microorganism most commonly used for vanillin bioconversion [12,33,34]). The production of bio-vanillin was carried out in 2 L bioreactors (BioStat B-plus, Sartorius, Germany). At the end of production (set for maximum vanillin production), the bioconversion broth was centrifuged at 4424× *g* for 20 min at 7 °C to remove the cells. The pH of the supernatant of the bioconversion broth was adjusted with HCl to a pH value to be defined in Section 3.1. Table 2 shows the concentration of phenolic compounds that may be present in the vanillin bioconversion broth (vanillin, ferulic acid, vanillic acid, vanillyl alcohol and guaiacol), and their saturation vapour pressures (p_vi_) and activity coefficients at infinite dilution ($\gamma_{i}^{\infty }$*)* at 25 °C. Based on HPLC analyses, no ferulic acid, vanillyl alcohol or guaiacol were detected in the broth. Only vanillin and vanillic acid were detected in the broth, within the limit of detection of each compound in the HPLC method used.

The standards for HPLC analyses were vanillin (99%, Sigma Aldrich, St. Louis, MO, USA), ferulic acid (≥99%, Sigma Aldrich, St. Louis, MO, USA), vanillic acid (≥97% Sigma Aldrich, St. Louis, MO, USA), vanillyl alcohol, (≥97% Sigma Aldrich, St. Louis, MO, USA), guaiacol (98%, Sigma Aldrich, St. Louis, MO, USA).

### 2.2. Experiments

#### 2.2.1. Pervaporation Fractionated Condensation Experiments

The pervaporation experimental set-up (Figure 1) consists of a closed feed vessel, a flat circular pervaporation module (GKSS, Germany) with an effective membrane area of 0.01 m^2^, two condensers-in-series and a single-stage rotary vane vacuum pump (Pfeiffer Duo 2.5, Pfeiffer Vacuum, Germany).

About 300 g of feed (Bioconversion broth 1, Table 2) was used in the pervaporation experiment. The feed was placed in a closed jacket feed vessel with a small headspace to minimise aroma losses to the gas phase. The jacket feed vessel was stirred at 100 rpm and the temperature of the feed, T_feed_ (°C), was kept constant using a heating and stirring plate. A recirculation pump (IMS 901B, Ismatec, Germany) drove the feed at a feed flow rate of 80 L·h^−1^ (measured at room temperature) into the pervaporation module, corresponding to an average Reynolds number in the feed compartment at room temperature of 430 [19,40].

The temperature of the feed stream, T_feed_ (°C), was set at 75 °C, a relatively high value of temperature, in order to maximise the flux of vanillin, but still lower than the maximum temperature of operation of the membrane (80 °C).

The condensers used in the downstream side of the pervaporation set-up were glass U-shaped traps. The first condenser was immersed in a refrigerated bath (Corio CP-300F, Julabo, Seelbach, Germany) and the second condenser in liquid nitrogen (−196 °C). The temperature of the first condenser was set at pre-defined values.

In order to avoid vanillin condensation in the circuit between the membrane module and the first condenser, this circuit was heated using an electric heating tape connected to a temperature controller (CB100, from RKC Instruments Inc., Tokyo, Japan), at 60 °C, as described in Brazinha et al. [19]. The permeate pressure was defined around 1.5 mbar, a value optimised by Brazinha et al. [19] in order to obtain maximised vanillin fluxes. Each experiment was performed for 4 h. Before each experiment, the membranes were conditioned with water for one hour.

#### 2.2.2. Vacuum Distillation Fractionated Condensation Experiments

The vacuum distillation set-up (Figure 2) consists of a closed feed vessel, three condensers-in-series, and a single-stage rotary vane vacuum pump (Pfeiffer Duo 2.5, Pfeiffer Vacuum, Asslar, Germany).

An amount of 5.53 ± 0.14 g of feed (Bioconversion broths 1 and 2, Table 2), was used in the vacuum evaporation experiments. The feed was stirred at 100 rpm and heated at different defined temperatures using a heating and stirring plate. Apart from the closed feed vessel, the vacuum distillation circuit was similar to the downstream side of the pervaporation set-up. The difference was the use of an additional condenser; the first two condensers were set at the same temperature, controlled by a refrigerated bath (Corio CP-300F, Julabo, Germany). The last condenser was identical to the pervaporation set-up (a trap immersed in liquid nitrogen). The vacuum for the whole set-up decreased very rapidly (within around 5 min) to pressure values below 1 mbar and, after 10 min, reached a value around 0.15 mbar. A summary of the operating conditions, which were varied during the vacuum distillation experiments performed in this work, are presented in Table 3.

Water was added to the feed vessel in the last 3 vacuum distillation experiments, in order to promote vanillin evaporation, keeping constant its high activity coefficient and hence its volatility. The water pulses were added every 15 min.

### 2.3. Methods for Estimating Relevant Thermodynamic Parameters

The saturation vapour pressure of water was estimated at 75 °C (the feed temperature used in the pervaporation experiment) and 85 °C (feed temperature used in vacuum distillation experiments), as shown in Table 4. The estimated saturation vapour pressure of water at different temperature values was calculated using the Antoine equation, for a temperature range between 0 °C and 100 °C [41].
(1)$$log_{10}\left(p\right)=8.0731-\frac{1730.63}{233.426+T}\;\;\;(p\;\mathrm{in}\;\mathrm{mmHg},\;T\;\mathrm{in}\;{}^{\circ}C)$$

The saturation vapour pressures and the activity coefficients of vanillin (the most abundant phenolic compound present in the vanillin bioconversion broth) were estimated at 25 °C (room temperature), as shown in Table 2, and at 75 °C and 85 °C, as shown in Table 4. The saturation vapour pressure of vanillin at 25 °C was estimated through Equation (2) [42,43] for a temperature range between 24 °C and 55 °C.
(2)$$log_{10}\left(p\right)=10.93562-\frac{4535.023}{T}\;(p\;\mathrm{in}\;\mathrm{kPa},\;T\;\mathrm{in}\;K)$$

The saturated vapour pressures of vanillin at 75 °C and 85 °C were estimated through the (3) [44], for a temperature range between 82 °C and 504 °C.
(3)$$log_{10}\left(p\right)=7.81755-\frac{2260.06}{233.426+T}\;\;\;(p\;\mathrm{in}\;\mathrm{mmHg},\;T\;\mathrm{in}\;{}^{\circ}C)$$

The activity coefficients (in aqueous solution) of vanillin were estimated using the mutual solubility method [45]. This method works only for very hydrophobic compounds, such as vanillin, and it is based on the measurement of the concentration of the aroma compound in an aqueous phase, equilibrated with the organic compound phase (vanillin). Indeed, if the thermodynamic equilibrium is established, the aroma compound activity in the organic phase and in the aqueous phase can be related as [45]:(4a)$$a_{i}^{aqu}=a_{i}^{org}$$
(4b)$$\gamma_{i}^{aqu}x_{i}^{aqu}=\gamma_{i}^{org}x_{i}^{org}$$

Assuming that the organic phase is almost pure compound (vanillin in this case), then
(4c)$$\gamma_{i}^{org}≅1\;\;and\;x_{i}^{org}≅1\;;\;\gamma_{i}^{\infty }=\frac{1}{x_{i}^{aqu}}\;\;\;\;\;\;\;\;\;\;\;\;\;\;\;$$
where *a_i_* is the chemical activity of compound *i*, *x_i_* is the molar fraction of compound *i* in a liquid phase and $\gamma_{i}^{\infty }$ is the activity coefficient at infinite dilution.

Where $x_{i}^{aqu}$ of vanillin at 25 °C is 1.23 × 10^−3^ [46] and at 75 °C it was calculated based on the concentration of vanillin in a saturated aqueous phase at this temperature [47]. The $x_{i}^{aqu}$ of vanillyl alcohol at 25 °C was calculated based on the concentration of vanillyl alcohol in a saturated aqueous phase, which is 16.12 g·L^−1^ [48].

The estimated saturated vapour pressure of guaiacol at 25 °C was calculated through the Antoine Equation (5) [44] for a temperature range between 31 °C and 424 °C.
(5)$$\log_{10}\left(p\right)=7.899-\frac{2204.06}{233.426+T}\;\;\;(p\;\mathrm{in}\;\mathrm{mmHg},\;T\;\mathrm{in}\;{}^{\circ}C)$$

### 2.4. Analytical Methods

Quantification of Phenolic Compounds

At the end of each pervaporation and vacuum distillation experiment, each condensate was recovered from the traps and was chemically characterised in terms of the concentration of phenolic compounds. When the condensates were in the solid form, the compounds present were recovered by adding hot water.

The concentration of the phenolic compounds (vanillin, vanillic acid, ferulic acid, vanillyl alcohol and guaiacol) was measured using HPLC, high-performance liquid chromatography. A high performance liquid chromatograph (Alliance e2695, Waters, Milford, MA, USA) was used, equipped with a UV detector at 280 nm and a Nova-Pak^®^ C18 3.9 mm × 150 mm column (particle diameter of 4 mm), preceded by a guard column. A gradient elution program at the flow rate of 0.5 mL.min^−1^ was used at 30 °C. The mobile phase was a mixture of two eluents, one of them (A) containing 10% (*v*/*v*) methanol, 2% (*v*/*v*) acetic acid in Milli-Q water and the other (B) containing 90% methanol, 2% acetic acid in Milli-Q water. The gradient elution program was 0% B (0–10 min), from 0 to 15% B (10–25 min), from 15 to 50% B (25–35 min) and 50 to 0% B (35–38 min).

A nuclear magnetic resonance (NMR) spectroscopy analysis was also performed. The 1H NMR spectra were obtained by dissolving the sample in deuterium oxide (D2O, 99.9%) and subjecting the samples to 3 freeze-drying cycles, with samples resuspended in deuterium oxide between those cycles. The samples were analysed using an Avance III 400 NMR spectrophotometer (Bruker, Fällanden, Switzerland), at 25 °C, with an NMR magnet operating at 400.13 MHz for 1H.

## 3. Results and Discussion

### 3.1. Optimisation of Feed pH Used in the Pervaporation and Vacuum Distillation Processes

The volatility of a dilute solute i in solution is quantified by the Henry constant of compound i, H_i_ (mbar), which can be calculated by the product of its saturation vapour pressure, pv_i_ (mbar), and its activity coefficient, $\gamma_{i}^{\infty }$ (-), at a given temperature. Table 5 compiles the values of the Henry constant, H_i_ (mbar) and the pKa_i_ of the compounds commonly present in a bioconversion broth for the production of vanillin at 25 °C.

Volatile compounds with high Henry constants, H_i_ (mbar), in non-ionic form (when the pH value of the solution is clearly lower than the pKa of the compound) may be recovered in the downstream side of the membrane in a pervaporation process. They can also be recovered by distillation at atmospheric or reduced pressure (under vacuum). On the other hand, non-volatile compounds, with extremely low Henry constant, H_i_ (mbar), and/or compounds that are in ionic form at the pH of the solution, [19] cannot be recovered by processes that require a minimum vapour pressure, such as pervaporation and distillation.

A complex vanillin bioconversion broth comprises various non-volatile compounds (glucose, salts, yeast extract [19]) that remain in the bioconversion broth. Other contaminants reported to be present in the vanillin bioconversion broth are noted in Table 2 and present a measurable volatility.

To assure the separation of vanillin from ferulic acid and vanillic acid, the pH value of the bioconversion broth has to be adjusted, in order to assure that vanillin is protonated (HVan) (which is the second most volatile compound, with the second highest Henry constant, see Table 5) and both ferulic acid (HFer^−^ and Fer^2−^) and vanillic acid (HVac^−^ and Vac^2−^) remain non-protonated (and, consequently, non-volatile). Camera-Roda et al., 2014 [26] proved that the performance of pervaporation for recovering/permeating vanillin is pH-dependent; permeation decreases at pH 7.9 when vanillin is partially dissociated and declines significantly at pH 10.3 when vanillin is completely dissociated [26].

Table 6 illustrates the relative fraction of the protonated form of vanillin (HVan), ferulic acid (H_2_Fer), vanillic acid (H_2_Vac), alcohol vanillyl (Halc) and guaiacol (HGuai) at different pH values, including at pH 8.2 (the pH value at the end of the vanillin bioconversion [51]) and at pH 7.2 (the pH value reported in [19] as optimal for vanillin bioconversion).

Table 6 shows that at pH 8.2, only 13.7% of vanillin is in protonated form, which has a negative effect on permeation if a pervaporation process is used [26]; the ferulic acid and vanillic acid that remain in the bioconversion broth are in their non-volatile, non-protonated (ionic) form. Therefore, a pH of 6.8 was selected because, at this pH, 81.7% of vanillin is in the protonated form and the ferulic acid and vanillic acid are still, mostly, in their non-protonated form (>99%) (Table 6), which makes vanillin recovery possible free from non-volatile ferulic acid and vanillic acid. Neither ferulic acid nor vanillic acid are expected to be recovered, so they will not contaminate the vanillin aimed product. The reason why a lower pH value was not selected results from the fact that the gain in terms of the fraction of protonated vanillin is not high if the pH is lowered from 6.8 to 6.5 and, on the other hand, the expenditure in acid to lower the pH of the bioconversion broth is higher.

Vanillyl alcohol is more volatile than vanillic acid and guaiacol is the most volatile of the compounds present (see Table 5), but none of these compounds was detected by HPLC (Table 2), probably because they were not produced during the bioconversion. So, although both guaiacol and vanillyl alcohol are protonated at the same pH values as vanillin, they will not contaminate the final product.

Although it is expected that vanillin will be recovered free from contaminants, a NMR analysis should be performed for the selected condensate to confirm whether or not the recovered vanillin is free from contaminants, particularly if vanillic acid, which is present in the bioconversion broth, is recovered.

### 3.2. Pervaporation-Fractionated Condensation Experiments

#### 3.2.1. Pervaporation

PDMS membranes have been applied for removing organics from aqueous mixtures (e.g., ethanol [52,53], methanol [52], phenol [54], methyl acetate [55], acetone [56]), exhibiting high selectivity and permeability to organic substances [31]. A PDMS membrane was also used to recovery apple juice aroma compounds [57]. To the best of our knowledge, no studies have reported the recovery of vanillin by pervaporation using PDMS membranes. Performing pervaporation for vanillin recovery from a bioconversion broth at a temperature of 75 °C makes sense from a process point of view (below the maximum temperature of operation of PDMS of 80 °C), because the vanillin flux is expected to increase significantly.

In the pervaporation process for vanillin recovery, from the liquid feed to the permeate vapor (before fractionated condensation), the parameters of permeability to vanillin, *P_van_* (mol·(m·s·Pa)^−1^) and selectivity of vanillin against water (-), allow a better understanding of the intrinsic performance of the membrane, and these parameters are adequate for comparing the transport properties of different membranes [58,59,60]. The permeability, *Pi*, to compound *i* (vanillin or water) is calculated by Equation (6a)
(6a)$$J_{i}=\frac{P_{i}}{\delta }(p_{i,feed}-p_{i,perm})$$
(6b)$$J_{i}=\frac{P_{i}}{\delta }\left(x_{i,feed\;\;}H_{i}-y_{i}\;p_{perm}\right)$$
where *J_i_* (mol m^−2^ s^−1^) is the molar flux, *p_i_* (Pa) is the partial pressure of compound *i* (feed refers to the feed side and perm refers to the permeate side), *x_i_* (-) is the molar fraction in the feed side, *y_i_* (-) is the molar fraction of compound *i* in the permeate side, *δ* (m) is the thickness of the dense top layer of the composite membrane and *p_perm_* (Pa) is the permeate total pressure.

The selectivity of vanillin against water is calculated by Equation (7) as the ratio of permeability between vanillin and water, and the separation factor of vanillin–water is calculated by Equation (8)
(7)$${\mathrm{Selectivity}}_{\mathrm{van}-\mathrm{water},\mathrm{pervaporation}}=\frac{p_{\mathrm{van}}}{p_{\mathrm{water}}}\;$$
(8)$$\mathrm{Separation}\;\mathrm{factor}=\frac{y_{\mathrm{van}}/y_{\mathrm{water}}}{x_{\mathrm{van}}/x_{\mathrm{water}}}$$

According to the study reported in Brazinha et al. [19] for maximised vanillin fluxes, a permeate pressure of 1.5 mbar should be employed for maximising the driving force.

Therefore, the permeate pressure value used was 1.5 mbar in the experiment using a PDMS membrane at 75 °C. The permeate pressure values were at the target value of 1.5 mbar. Table 7 shows the Henry constants of compound *i* (vanillin and water) at 75 °C.

The different parameters related to the pervaporation process are shown in Table 8.

#### 3.2.2. Pervaporation with Fractionated Condensation

Bio-vanillin recovery from a bioconversion broth using a pervaporation-fractionated condensation process, with two condensers-in-series at 1.5 mbar permeate pressure, with the first condenser at 0 °C and the second condenser at −196 °C, has been reported to lead to a separation factor of vanillin in relation to water of an infinite value (the recovery of pure vanillin from a bioconversion broth) [19]. The condensation of vanillin, which occurred at the first condenser [19], was due to the very different condensation behaviour of vanillin and water, reflected in their very different saturation vapor pressure values (see Table 4).

In this work, the aim is to maximise the vanillin flux and the selectivity of vanillin against water and provide conditions that assure the recovery of vanillin from the bioconversion broth free from contaminants. A pervaporation-fractionated condensation experiment was performed, using a PDMS membrane at 75 °C.

The values of permeate pressure and temperature of the first condenser was previously optimised [19]. The value of vacuum pressure selected for the pervaporation-fractionated condensation system was 1.5 mbar, considering the best balance between the required driving force for the permeation of vanillin and the energy input associated, and the average value of permeate pressure for all pervaporation experiments performed. The defined temperature of the first condenser was 0 °C. The rest of the operating conditions are defined in Section 2.2.1.

The results of the pervaporation-fractionated condensation experiment are presented in Table 9, where it is shown the total flux of vanillin and the percentage of the total permeated vanillin compounds recovered in the different sections of the permeate circuit (first and second condensers and permeate tubes). In the first condenser, vanillin was recovered in the solid form without water (vanillin was collected by rinsing with hot water). The permeate tubes were also rinsed with hot water, to assure that all vanillin attached to the tubes was recovered. The % of vanillin recovered in each section was performed by calculating the mass of vanillin that permeated and comparing it with the total mass of the compound recovered in the different sections of the downstream circuit (the two condensers and the permeate tubing). Table 9 also presents the mass balance for vanillin. The mass balance deviation was calculated by the mass of vanillin in the initial feed and comparing it with the sum of the total mass of vanillin recovered in the different sections of the downstream circuit and remaining in the feed at the end of each experiment. Figure 3 shows the HPLC chromatograms obtained, referring to samples taken during the experiment using a PDMS membrane at 75 °C.

Vanillin is the only compound recovered in the permeate circuit, and vanillic acid was not detected in any sample from the permeate circuit (the chromatograms are similar for all permeate samples). Therefore, the vanillic acid fluxes can be considered to be null or close to zero, considering the detection limit of the HPLC method used. The HPLC analyses show that vanillic acid and other potential contaminants (Table 2) are not detected in the feed stream nor in the condensers.

The vanillin recovered in the first condenser was in solid form, without water, so the separation factor of vanillin in relation to water has an infinite value. However, only about 46% of the permeated vanillin is recovered in the first condenser (Table 9). The remaining permeated vanillin is recovered in the second condenser together with water (Table 9). However, it should be stated that at 0 °C, all vanillin was thermodynamically expected to be retained in solid form in the first condenser, due to its extremely low saturation vapor pressure values.

This deviation can be explained by the fact that the flux of water is much higher than the flux of vanillin (see Table 8), which causes vanillin to be dragged with water to the second condenser. In order to increase the amount of solid vanillin in the first condenser, the residence time/area of the first condenser should be increased and/or the temperature of the first condenser should be decreased. In the case of a temperature decrease in the first condenser, the values could be lowered to between −5 and −10 °C (higher than −13 °C, the saturation vapour pressure of water at 2 mbar). However, increasing the area of the first condenser is probably a more economic strategy that should be implemented and tested first, preferably in a pilot-scale study.

### 3.3. Vacuum Distillation-Fractionated Condensation Experiments

The pervaporation experiments, without fractionated condensation, have shown a relatively good selectivity for the recovery of vanillin. The selectivity of vanillin against water was 4.5 when using a PDMS membrane with a feed temperature of 75 °C. The selectivity of vacuum distillation (-), a purely evaporative process, was calculated for the separation of vanillin from water, using the following equation:(9)$${\mathrm{selectivity}}_{\mathrm{van}-\mathrm{water},\mathrm{evaporative}}=\frac{H_{\mathrm{van}}}{H_{\mathrm{water}}}\;$$

The value for the selectivity of vanillin against water (-) obtained by vacuum distillation is 0.03, at a feed temperature of 75 °C. This value is substantially lower than the value for selectivity in pervaporation, which was expected due to the presence of a selective hydrophobic membrane top layer with affinity to vanillin. On the other hand, the selectivity of water against vanillin (-) has the value of 35.31 at 75 °C, meaning that water is much more volatile than vanillin in this process. In order to recover vanillin, vacuum distillation needs to be combined with fractionated condensation.

A set of experiments was performed to recover vanillin through vacuum distillation with fractionated condensation (using three condensers-in-series, see Figure 2), in order to increase the condensation area, aiming at condensing as much as possible of the evaporated vanillin. The objective was to achieve the evaporation of both water and vanillin (avoiding non-volatile potential contaminants) and then promote the condensation of vanillin in the two first condensers. As water is much more volatile than vanillin, the Henry constant of water is twice the order of magnitude of vanillin (Table 7), it is relatively easy to separate vanillin from water in the first two condensers. Table 10 shows the results of the evaporation of vanillin during the vacuum distillation experiments.

The duration of the experiment from 15 min to 70 min (Experiments 1 and 2) did not lead to an increase in vanillin evaporation, meaning that vanillin evaporated during the first few minutes of each experiment. Experiments 1 and 2 show a low percentage of evaporated vanillin. This behaviour can be explained by the fact that water evaporates much faster than vanillin. After evaporation of water, vanillin becomes more concentrated (and consequently with a lower activity coefficient in the concentrated aqueous solution), which lowers its volatility in solution. Therefore, in order to increase the rate of evaporation of vanillin without increasing the temperature, a strategy of multiple water feeding pulses was implemented. This approach consists of adding several consecutive water pulses, each one with the same mass of the evaporated water, keeping constant the total mass of the feed solution. As shown in Table 10, in Experiments 3–5, adding more pulses of water to the remaining broth increased the recovery of vanillin, making possible to achieve a total vanillin recovery of 88% (Experiment 5) after the addition of 18 water pulses with the same mass each. In each experiment, the mass balance of vanillin closed with a slight negative value, most likely due to some vanillin losses in the vacuum distillation circuit, but deviations were small and reasonable (≤10%). Regarding the flux of vanillin in the vacuum distillation process, although the mass of evaporated vanillin increased with an increasing number of water pulses, the overall flux of vanillin decreased slightly from 3.7 g·m^−2^ h^−1^ with 10 water pulses to 2.7 g·m^−2^ h^−1^ with 18 water pulses.

From the set of vacuum distillation experiments performed with the same bioconversion broth 2 (see Table 10), Experiment 5 was selected as it corresponds to the highest percentage of recovered vanillin, 87.8 ± 5.8%, as well as to low mass balance deviation of vanillin (−5.6%) and no vanillic acid in the condensates.

An additional vacuum distillation experiment, Experiment 6, was performed with the same operating conditions as the selected vacuum distillation experiment (Experiment 5) but instead using the bioconversion broth 1, which was used in the pervaporation experiments. As expected, Experiments 5 and 6 performed similarly (see Table 10).

Figure 4 shows the HPLC chromatograms obtained. The chromatograms were taken at the end of Experiment 5 (the same as in Experiments 1–4, where the bioconversion broth 2 was processed) and at the end of Experiment 6 (where the bioconversion broth 1 was processed), this can be seen in Table 10.

As, in pervaporation experiments, vanillin and vanillic acid were present in the feed and no vanillic acid was detected in the condensers (see Figure 4(2b–2d)). In the first and second condensers, vanillin was obtained pure in solid form without water. However, as shown in Figure 4(2d)), vanillin is still present in the third condenser, indicating that vanillin was dragged with water, as it seems to happen in the pervaporation experiments. In order to decrease the amount of vanillin lost to the final total vapour condenser (together with water), the area of the first condenser should be increased

The chromatograms from Experiment 5 are similar to the ones from Experiment 6, except for vanillyl alcohol. Indeed, a peak of vanillyl alcohol was detected by HPLC in the final feed (see Figure 4(1a)), indicating the presence of vanillyl alcohol in the feed. Consequently, a very small peak of vanillyl alcohol was also detected in the condensers in Experiment 5 (Figure 4(1b–1d)), due to the presence of a minor amount of vanillyl alcohol in the initial feed.

The condensate from vacuum distillation Experiment 5 (condensate obtained in the first condenser), the final product of the experiment, was qualitatively characterised by 1H-NMR to confirm the presence of vanillyl alcohol and to assess the presence of other contaminants. The sample was completely saturated in order to observe the potential presence of compounds with a low concentration. Figure 5 represents the 1H-NMR spectrum, for the first condensate of vacuum distillation Experiment 5.

As shown in Figure 5, vanillin is present in the sample, since the proton characteristic of the aldehyde linkage, -CHO (linkage A in the Figure 5), is represented at 9.51 ppm. This linkage is not present in the other contaminants (ferulic acid, vanillic acid, alcohol vanillyl and guaiacol). On the 1H-NMR spectrum, a peak was also observed at 4.43 ppm, characterised by the protons of the linkage -CH_2_- (linkage G in the Figure 5). This linkage is only present in alcohol vanillyl, it is not present in ferulic acid, vanillin, vanillic acid or guaiacol, confirming the presence of alcohol vanillyl in the sample. The proton of the linkage -OH in both compounds (linkage E in Figure 5) is not detected on the ^1^H-NMR spectrum using the D_2_O solvent [61]. Through the integration of the peaks, it was possible to confirm that only vanillin and vanillyl alcohol were present in the sample, since there were no remaining protons.

In terms of the total composition of the different streams at the end of each experiment, both Experiments 5 and 6 performed similarly (see Figure 4) because the initial feed contained vanillyl alcohol in minor quantities, which was not detected by HPLC but was confirmed by NMR analysis of the condensate. However, vanillyl alcohol is also present in botanical vanillin [62]. Rhodia (Solvay) produces a bio-vanillin containing 3% by weight of impurities, where the most abundant impurity is vanillyl alcohol [30].

When comparing the selected pervaporation-fractionated condensation experiment (with a PDMS membrane at 75 °C) to the selected vacuum distillation-fractionated condensation experiment, at 85 °C, with the addition of several consecutive water pulses (Experiments 5–6), we may conclude that both integrated processes allow for recovering pure vanillin free from undesirable contaminants. These processes should be compared in terms of process productivity. The selected pervaporation-fractionated condensation and the selected vacuum distillation-fractionated condensation processes exhibited vanillin fluxes, J_van_ (g·m^−2^ h^−1^), respectively, of 1.19 ± 0.01 g·m^−2^ h^−1^ (Table 9) and 2.7 ± 0.1 g·m^−2^ h^−1^ (Table 10).

Both processes should also be compared in terms of energy consumption per mass of vanillin recovered in units of KJ kg_VAN_^−1^, which is calculated by Equation (10), considering, for the sake of simplicity, that water is the only species that contributes to the energy consumption:(10)$$E\;\left({\mathrm{kJ}\;\mathrm{kg}}_{\mathrm{VAN}}^{-1}\right)\sim \frac{m_{\mathrm{water}\;}[\left({\mathrm{Cp}}_{\mathrm{water}}\times \left({\;T}_{\mathrm{final}}-T_{\mathrm{initial}}\right)+\Delta H_{\mathrm{vaporization}.\mathrm{water}}\left(T_{\mathrm{final}}\right)\right]}{m_{\mathrm{VAN}}}$$
where *m_water_* (kg) and *m_VAN_* (kg) are, respectively, the mass of water and the mass of vanillin permeated (in the pervaporation-fractionated condensation experiment) or evaporated (in the vacuum distillation-fractionated condensation experiment). For this calculation, the values obtained experimentally were used: total flux of 2.38 g·m^−2^ h^−1^ and 10.76 g·m^−2^ h^−1^, respectively, and a vanillin flux of 1.19 × 10^−3^ g·m^−2^ h^−1^ and 2.7 × 10^−3^ g·m^−2^ h^−1^, respectively. The other values in the equation have the following definitions: *Cp_water_* (kJ·kg^−1^·K^−1^) is the specific heat capacity of water of 4.2 J^−1^ (g K); *T_initial_* is the temperature of bioconversion of 45 °C; *T_final_* is the operating temperatures of pervaporation and vacuum distillation, 75 °C and 85 °C, respectively, and Δ*H_vaporisation.water_* (*T_final_*) is the enthalpy of *vaporisation* of *water* at the operating temperature (2320.6 J/g at 75 °C for pervaporation-fractionated condensation and 2295.3 J^−1^ g at 85 °C for vacuum distillation).

As a result of this calculation, we may conclude that the thermodynamic energy necessary to produce a *Kg* of vanillin is 1361 KWh, which translates into a cost of 136.1 EUR/Kg_vanilina_^−1^ for the pervaporation-fractionated condensation, and 2727 KWh, which translates into a cost of 272.7 EUR/Kg_vanilina_^−1^ for vacuum distillation-fractionated condensation, considering an energy cost of 0.10 EUR/KWh^−1^. Pervaporation requires a lower energy input because, due to its high selectivity for vanillin against water, it involves a lower flux of water for the same vanillin recovered. As a consequence, the energy input required is lower when the pervaporation process is used.

Also in pervaporation, the energy consumption of the vacuum pump is rather small, and may be considered negligible, when compared with the energy for evaporation/pervaporation. Indeed, after setting the vacuum pump at 1.5 mbar, this pressure remains almost constant without pump work because the water and vanillin fluxes are rather mild and non-condensable gases are not present and, therefore, there is no need for pump work to remove them.

Considering that the market value of biovanillin is around USD/ kg^−1^ 400–600 [63], it is clear from this study that the selection of the recovery process is extremely important in order to assure a positive return of investment.

## 4. Conclusions

A selected pervaporation-fractionated condensation study was performed with a PDMS membrane at 75 °C and compared with a vacuum distillation-fractionated condensation process performed at 85 °C. Both processes allow for the recovery of vanillin without the presence of undesirable contaminants, if a pH correction to the final bioconversion medium is performed. These two processes were compared in terms of vanillin flux, which, as expected, is higher for the vacuum distillation process (2.7 ± 0.1 g·m^−2^ h^−1^ for vacuum distillation against 1.19 ± 0.01 g·m^−2^ h^−1^ for pervaporation). This result was expected because the membrane allows for a higher selectivity but introduces an additional barrier to transport. Consequently, in the pervaporation-fractionated condensation process, the pervaporation step shows a selectivity for vanillin against water of 4.5 (acting as a preconcentration step). One of the most important aspect is the required energy involved in both processes, which was calculated on the basis of a thermodynamic balance; the vacuum distillation process requires 2727 KWh kg_VAN_^−1^ at 85 °C (corresponding to 272.7 EUR/kg_VAN_^−1^), while the pervaporation process requires only 1361 KWh kg_VAN_^−1^ at 75 °C (corresponding to 136.1 EUR/kg_VAN_^−1^). It should be emphasised that, in these calculations, the energy involved in the condensation process, which is significantly higher in the vacuum distillation process where the water flux is significantly higher (less selective process for vanillin), was not considered. Future work at pilot scale requires the design of condensers with optimised area/residence time of condensable compounds, in order to assure a high recovery of the permeated vanillin. These pilot 2studies should be performed for both approaches discussed in this work. Considering the actual energy cost of EUR/KWh^−1^ 0.10 for industrial production and the current risk of increasing energy price, the pervaporation-fractionated condensation approach is recommended. It should be also emphasized that this approach does not involve the use of solvents with inherent costs of solvent reuse/disposal.

## Figures and Tables

**Figure 1 membranes-12-00801-f001:**
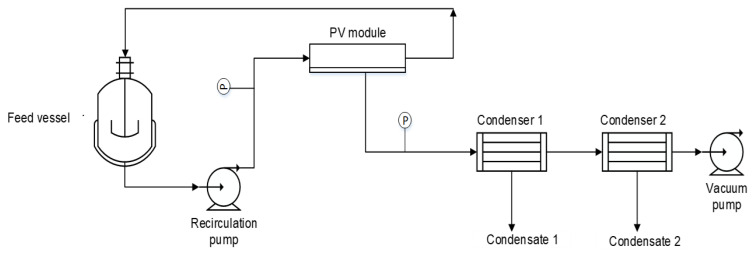
Experimental pervaporation-fractionated condensation laboratory set-up similar to [19]. P is a pressure transducer.

**Figure 2 membranes-12-00801-f002:**
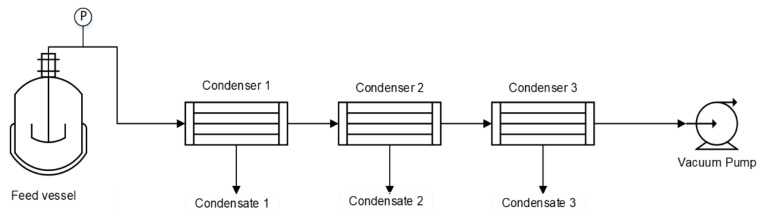
Experimental vacuum distillation-fractionated condensation laboratory set-up. p is the pressure transducer.

**Figure 3 membranes-12-00801-f003:**
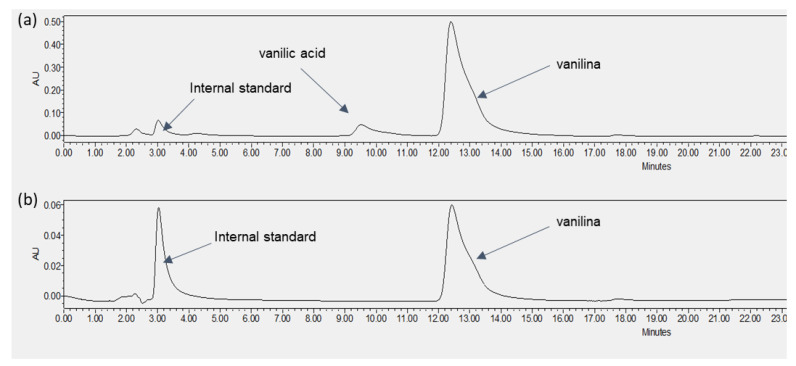
Pervaporation-fractionated condensation experiment using a PDMS membrane at a feed temperature of 75 °C and a first condenser at 0 °C. HPLC chromatograms refer to different samples taken during the experiment: (**a**) Final feed solution; (**b**) Condensate in the first condenser.

**Figure 4 membranes-12-00801-f004:**
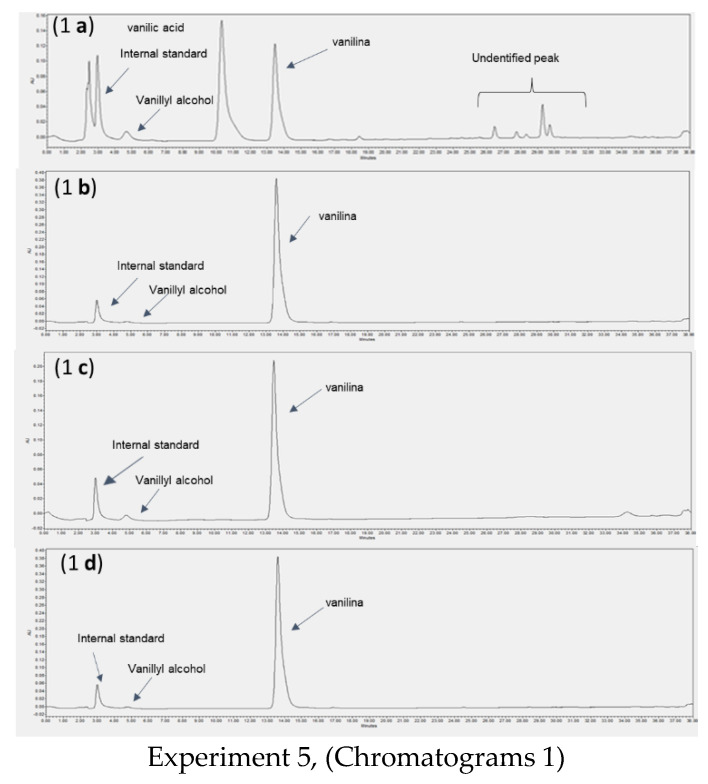
Vacuum distillation experiments: HPLC chromatograms obtained at the end of Experiment 5 (Chromatograms 1) and at the end of Experiment 6 (Chromatograms 2). For each experiment, different samples were analysed: (**ia**) final feed; (**ib**) 1st condenser; (**ic**) 2nd condenser; and (**id**) 3rd condenser, where i is the number of the chromatogram.

**Figure 5 membranes-12-00801-f005:**
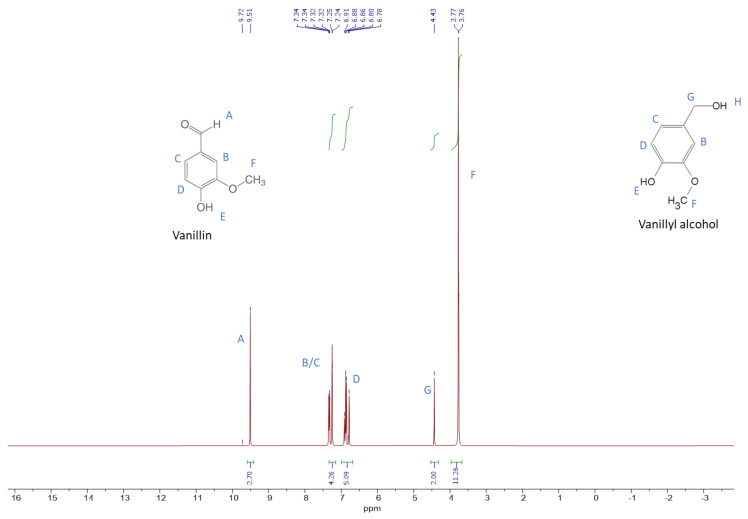
The 1H-NMR spectrum of the first condenser of vacuum distillation Experiment 5. Note: Letters identify the different types of protons present in vanillin and vanillyl alcohol molecules.

**Table 1 membranes-12-00801-t001:** Main characteristics of the pervaporation membrane used in this work (information provided by the manufacturer).

Membrane	Manufacturer	Material of the Active Layer	T_maximum_ ^a^ (°C)	pH (–)	δ_active.layer_ ^b^ (µm)
Pervap^TM^ 4060	Deltamem, Switzerland	Polydimethyl siloxane (PDMS)	80	5–8	~2

Legend: ^a^ T_maximum_, maximum temperature of operation; δ_active layer_ ^b^, thickness of the active layer of the membrane.

**Table 2 membranes-12-00801-t002:** Concentration of phenolic compounds present in the vanillin bioconversion broth (the feed for this work) and their saturation vapour pressures (p_vi_) and activity coefficients ($\gamma_{i}^{\infty }$ ) at 25 °C.

Compound	Concentration (g·L^−1^)	p_vi_ (mbar) (At 25 °C)	$\gamma_{i}^{\infty }\;(-)$ (At 25 °C)
Vanillin	3.99 ± 0.02 ^(1)^, 4.81 ± 0.41 ^(2)^	3.1 × 10^−4 (4)^	813.0 ^(4)^
Ferulic acid	<0.08 ^(1, 2)^	3.59 × 10^−6^ [35]	1.3 [36]
Vanillic acid	1.03 ± 0.04 ^(1)^, 1.48 ± 0.1 ^(2)^	(6.50 ± 0.98) × 10^−7^ [37]	2.8 [36]
Vanillyl alcohol	<0.12 ^(1–3)^	7.04 × 10^−6^ [38]	523.76 ^(4)^
Guaiacol	<0.10 ^(1–3)^	0.33 ^(4)^	273.7 [39]

^(1)^ Bioconversion broth 1; ^(2)^ Bioconversion broth 2; ^(3)^ Measured from real bioconversion media. Each value corresponds to the limit of detection for each compound in these samples (HPLC limit of detection); ^(4)^ See estimation of these parameters explained in Section 2.3 by Equation (4c).

**Table 3 membranes-12-00801-t003:** Operating conditions showing the operating values used during the vacuum distillation-fractionated condensation experiments.

Experiments	Feed Temperature T_feed_ (°C)	Addition of Water	Time of Experiment T_experiment_ (Min)
1 ^(a)^	85	No	15
2 ^(a)^	85	No	70
3 ^(a)^	85	Yes (4 pulses) ^(c)^	15 × 4 = 60
4 ^(a)^	85	Yes (10 pulses) ^(c)^	15 × 10 =150
5 ^(a)^	85	Yes (18 pulses) ^(c)^	15 × 18 = 270
6 ^(b)^	85	Yes (18 pulses) ^(c)^	15 × 18 = 270

^(a)^ Where the feed was the bioconversion broth 2 (Table 2); ^(b)^ Where the feed was the bioconversion broth 1 (Table 2); ^(c)^ Several consecutive water pulses were added, each with the same mass as the evaporated water (keeping constant the volume in the feed vessel).

**Table 4 membranes-12-00801-t004:** Values of the estimated vapour pressure p_vi_ (mbar) (**A**) and of the estimated activity coefficient at infinite dilution (in aqueous solutions) $\gamma_{i}^{\infty }$ (**B**).

**(A)**	**Vanillin**	**Water**
p_vi_ (mbar) (at 75 °C)	6.7 × 10^−2^	385.98
p_vi_ (mbar) (at 85 °C)	0.15	579.10
**(B)**	**Vanillin**	**Water**
$\gamma_{i}^{\infty }$ (-) (at 75 °C)	161.6	1.0
$\gamma_{i}^{\infty }$ (-) (at 85 °C)	-	1.0

**Table 5 membranes-12-00801-t005:** Henry constant and pKa of compounds commonly present in a vanillin bioconversion broth at 25 °C.

Compound	H_i_ (mbar)	pKa_i_ (-)
Vanillin	0.25	7.4 [49]
Ferulic acid	3.7 × 10^−4^	4.58 [49]
Vanillic acid	1.8 × 10^−6^	4.16 [50]
Vanillyl alcohol	3.69 × 10^−3^	9.92 [50]
Guaiacol	90.3	9.98 [50]

**Table 6 membranes-12-00801-t006:** Fraction of vanillin (HVan), ferulic acid (H_2_Fer), vanillic acid (H_2_Vac), vanillyl alcohol (Halc) and guaiacol (HGuai) in protonated form at different pH values.

pH	HVan	H_2_Fer	H_2_Vac	Halc	HGuai
6.5	88.5%	1.1%	0.6%	100.0%	100.0%
6.8	81.7%	0.6%	0.4%	100.0%	100.0%
7.2	61.3%	0.2%	0.1%	100.0%	100.0%
8.2	13.7%	0.0%	0.0%	98.1%	98.4%
10.3	0.0%	0.0%	0.0%	29.4%	32.4%

**Table 7 membranes-12-00801-t007:** Values of the calculated Henry constants of vanillin and water at 75 °C.

Compound	H_i_ (mbar) (75 °C)
Vanillin	10.89
Water	384.63

**Table 8 membranes-12-00801-t008:** Pervaporation operating and performance parameters of the membrane tested (PDMS, Pervap^TM4^060) at 75 °C, where J_i_, T is the total flux of the compound i and P_i_ is the permeability of compound i.

Membrane	PDMS
Temperature (°C)	75 °C
J_vanillin, T_ (g·m^−2^ h^−1^)	1.19 ± 0.01
J_water, T_ (Kg·m^−2^ h^−1^)	2.38
P_van_ (mol·(m·s·Pa)^−1^)	(8.71 ± 0.31) × 10^−12^
P_water_ (mol·(m·s·Pa)^−1^)	1.92 × 10^−12^
Selectivity_van-w_ (-)	4.5
Separation factor_van-w_ (-)	0.13

**Table 9 membranes-12-00801-t009:** Pervaporation-fractionated condensation experiment at 75 °C: total vanillin flux, percentage of permeated vanillin recovered in each section of the permeate circuit, in relation to the total mass of vanillin permeated and mass balance deviation to vanillin.

J_van_ (g·m^−2^ h^−1^)	1st Cond. (%)	2nd Cond. (%)	Permeate Tubes (%)	Mass Balance Deviation (%)
1.19 ± 0.01	44.5 ± 1.6	51.9 ± 1.8	3.5 ± 0.10	−3.0

Note: The feed was the bioconversion broth 1 (Table 2).

**Table 10 membranes-12-00801-t010:** Vacuum distillation with fractionated condensation: percentage of evaporated vanillin, mass balance deviation to vanillin and total vanillin flux.

Experiment	Temperature (°C)	Time (min)	Vanillin_evaporated_ ^(a)^ (wt%)	Vanillin Balance Mass Deviation (%) ^(d)^	J_vanillin, T_ (g·m^−2^ h^−1^)
1 ^(b)^	85	15	18.6 ± 1.3	−10.6	12.9 ± 0.6
2 ^(b)^	85	70	16.7 ± 1.3	−9.9	4.5 ± 0.1
3 ^(b)^	85	15 (x4)	42.5 ± 7.9	−9.3	6.7 ± 0.3
4 ^(b)^	85	15 (x10)	68.2 ± 4.5	−1.4	3.7 ± 0.2
**5 ^(b)^**	**85**	**15 (x18)**	**87.8 ± 5.8**	**−5.4**	**2.7 ± 0.1**
6 ^(c)^	85	15 (x18)	84.4 ± 0.46	−4.0	2.3 ± 0.0

^(a)^ Vanillin recovery in the 3 condensers, vanillin evaporated from the feed vessel; ^(b)^ where the feed was the bioconversion broth 2 (Table 2); ^(c)^ where the feed was the bioconversion broth 1 (Table 2); ^(d)^ a mass balance between the vanillin in the feed solution and in the sum of the vanillin in the outlets streams.

## Data Availability

Not applicable.

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
