# Peer review of "Comparative Analysis of Bio-Vanillin Recovery from Bioconversion Media Using Pervaporation and Vacuum Distillation"

_membranes, 2022, doi:10.3390/membranes12080801_

Round 1
Reviewer 1 Report
The present work discussed and compared the bio-vanillin recovery from bioconversion media using pervaporation and vacuum distillation, and the experiment processes are clear. The present work is interesting. I think it could be accepted after revision.
1. Be careful with the quality of the Figures
2. How about the art of the state of the present work? The author can provide more comparison with others work
3. Be careful with the format of references.
Author Response
Note: All the alterations introduced in the revised manuscript are highlighted in yellow.
The present work discussed and compared the bio-vanillin recovery from bioconversion media using pervaporation and vacuum distillation, and the experiment processes are clear. The present work is interesting. I think it could be accepted after revision.
- Be careful with the quality of the Figures
Figures 3 and 4 were modified.
2) How about the art of the state of the present work? The author can provide more comparison with others work
We reported an extensive list of examples taken from literature related to the recovery/purification of vanillin from complex bioconversion broths, namely:
- by pervaporation (Lines 73-88),
- by alternative technologies such as crystallisation, adsorption, liquid-liquid extraction and membrane-based solvent extraction (Lines 54-72) and
- by vacuum distillation (Lines 89-93)
3) Be careful with the format of references.
References were carefully reviewed and the errors found were rectified.

Reviewer 2 Report
Valerio et al. reported a comparative study of bio-vanillin recovery from bioconversion broths from a fractioned condensation process distinctly coupled to two different vanillin recovery/purification processes. They also well described experimental set-up and analytical and theoretical methods for both coupled processes, the pervaporation-fractionated condensation process and the vacuum distillation-condensation fractionation. The work is well-written and is of interest to the food and pharmaceutical industry. Overall, I recommend this article for publication with some major revisions:
1) in this work, the pervaporation process is adopted as a preconcentration step when coupled with condensation fractionation. I would like to suggest to authors to add data related to the pore size of used PDMS membrane;
2) there are some overlapping expressions in this paper. For example, the same sentence is repeated in the abstract (at lines 14-16) and in introduction as well (at lines 91-94). Maybe you can even split the sentence for a better understanding. In the abstract there are repeated statements in the Results and discussion part as well;
3) the authors can supplement a few more sentences on the limitation of the recent approaches on pilot scale design, because it is not clear which of the way forward in the Conclusion section. Which of the two processes coupled with condensation fractionation is preferable at pilot scale?
I have some suggestions for improvement of the manuscript in the attached file.

Author Response
Manuscript ID: membranes-1859515
Comparative analysis of bio-vanillin recovery from bioconversion media using pervaporation and vacuum distillation
Note: All the alterations introduced in the revised manuscript are highlighted in yellow.
Reviewer 2
Valerio et al. reported a comparative study of bio-vanillin recovery from bioconversion broths from a fractioned condensation process distinctly coupled to two different vanillin recovery/purification processes. They also well described experimental set-up and analytical and theoretical methods for both coupled processes, the pervaporation-fractionated condensation process and the vacuum distillation-condensation fractionation. The work is well-written and is of interest to the food and pharmaceutical industry. Overall, I recommend this article for publication with some major revisions:
- in this work, the pervaporation process is adopted as a preconcentration step when coupled with condensation fractionation. I would like to suggest to authors to add data related to the pore size of used PDMS membrane;
As requested by the reviewer, the following text was added:
- In lines 116-117 : “The PDMS membrane is dense, without permanent pores.”
- there are some overlapping expressions in this paper. For example, the same sentence is repeated in the abstract (at lines 14-16) and in introduction as well (at lines 91-94). Maybe you can even split the sentence for a better understanding. In the abstract there are repeated statements in the Results and discussion part as well;
Considering the reviewer’s comments, the following modifications were introduced:
- Abstract, lines 13-19: “This work discusses the recovery and fractionation of bio-vanillin by pervaporation and by vacuum distillation, coupled with condensation fractionation, from bioconversion broth, to re-cover vanillin, free of potential contaminants, with maximised fluxes and selectivity for vanillin against water and minimised energy consumption per mass of vanillin recovered. This work discusses the recovery and fractionation of bio-vanillin, from a bioconversion broth, by pervaporation and by vacuum distillation, coupled with condensation fractionation, from a bioconversion broth. The objective was to recover vanillin free from potential contaminants, with maximised fluxes and selectivity for vanillin against water and minimised energy consumption per mass of vanillin recovere
- We believe that the following text should be emphasised in the Introduction section:
In lines 94-102: “Both pervaporation and vacuum distillation technologies, combined with fractionated condensation, will be evaluated and compared in terms of the objective of this work (recovery of pure vanillin from bioconversion media, maximised vanillin fluxes and selectivity of vanillin against water and minimised energy consumption per mass of vanillin recovered). The main objective of this work is to evaluate and compare the use of pervapora-tion and vacuum distillation, both coupled with condensation fractionation, for the recovery of vanillin free of contaminants. from a complex bioconversion broth, with maximised vanillin fluxes, selectivity of vanillin against water and minimised energy consumption per mass of vanillin recovered.”
3) The authors can supplement a few more sentences on the limitation of the recent approaches on pilot scale design, because it is not clear which of the way forward in the Conclusion section. Which of the two processes coupled with condensation fractionation is preferable at pilot scale?
In the conclusion section, the different comparing parameters of the objective of this work were summarised:
- In lines 634-640: “Future work at pilot scale requires the design of condensers with optimised area/residence time of condensable compounds, in order to assure a high recovery of the permeated vanillin. These pilot studies should be performed for both approaches discussed in this work. Considering the actual energy cost of 0.10 € KWh-1 for industrial production and the current risk of increasing energy price, the pervaporation-fractionated condensation approach is recommended. It should be also emphasised that this approach does not involve the use of solvents with inherent costs of solvent reuse/disposal”.
I have some suggestions for improvement of the manuscript in the attached file.
Suggestions for authors_ membranes-1859515-peer-review-v1
- Please correct PEBA (polyether block amide) at line 69.
-At line 130, table 2 should report “Concentration of phenolic compounds that may be present in the vanillin bioconversion broth….” as indicated in the text at line 123.
Instead, the information added was in line 141: “measured from real bioconversion media”
-Table 2 is somewhat confusing. I suggest replacing the references (1) - (4) with letter (a) - (d) and checking the misdealing reference brackets.
- Please correct Tfeed with Tfeed at lines 152 and 156.
-Please replace the references (1)-(3) whit letters (a)-(c) for avoiding the misleading in Table 3.
- In the conclusions the vanillin flux values should be inverted at lines 587 -588.
-Please remove the extra point erroneously present in many units of measurement reported throughout the text (included tables), for example 2.7±0.1g.m-2h -1 and 1.19±0.01 g.m-2h -1 , at line 22 or 2727 KWh.kgVAN-1 and 1361 26 KWh.kgVAN-1 at lines 26-27.
- Please add (B) after (γi ∞) at the line 241
- Please correct pperm (Pa) at line 354
All the suggestions of the Reviewer 2 were incorporated in the manuscript, with one exception “at line 130”, now line 141, enhanced in yellow

Reviewer 3 Report
Comments on
Comparative analysis of bio-vanillin recovery from bioconversion media using pervaporation and vacuum distillation
In generals:
1. This paper investigated how to optimize a high vanillin recovery or flux with low energy consumption and no containments in the vanillin permeate from pervaporation and vacuum distillation. The considered parameters include pH in the bioconversion broth, vacuum pressure, energy consumption of vacuum pump, contaminates in vanillin permeate, and water addition in the feed. I suggest the author elaborate more on the priority list regarding the above-considered parameters.
2. Vacuum pressure affects the boiling point of the constituents in the bioconversion broth while the pH of the bioconversion broth affects the volatility of the constituents in the bioconversion broth. Vacuum pressure dominates the vanillin permeate flux while pH controls the containments of the vanillin permeate. The author needs to address more in adjusting the pH and vacuum pressure despite considering the energy consumption of the vacuum pump.
3. The author needs to explain why the addition of consecutive water pulses in the feed can promote more vanillin permeate yield. Although the addition of water in the feed may promote vanillin yield, the pH in the bioconversion broth may change. The author needs to prove no containments in the vanillin permeate after the water is added to the feed.
4. The normal practice in VOC separation from a mixture stream in some consecutive condensers is to condense water at the first condenser. In this study, only 46% vanillin was recovered in the first condenser. To separate vanillin from water The temperature setting of the first condenser is to separate volatile constituents of the bioconversion broth from water, the consecutive condenser then is used to separate vanillin from potential containments.
5. In this study, only the energy of vacuum pumps was compared. The author agrees that the energy consumption of the condensers is significant in vacuum vanillin recovery systems. To exclude the energy consumption of condensers from the investigated system is incomplete in the assessment of the system performance.
Author Response
Manuscript ID: membranes-1859515
Comparative analysis of bio-vanillin recovery from bioconversion media using pervaporation and vacuum distillation
Note: All the alterations introduced in the revised manuscript are highlighted in yellow.
Reviewer 3
- This paper investigated how to optimize a high vanillin recovery or flux with low energy consumption and no containments in the vanillin permeate from pervaporation and vacuum distillation. The considered parameters include pH in the bioconversion broth, vacuum pressure, energy consumption of vacuum pump, contaminates in vanillin permeate, and water addition in the feed.
I suggest the author elaborate more on the priority list regarding the above-considered parameters.
The pH of the bioconversion broth was previously optimised to 6.8 and kept constant, as discussed in section 3.1.:
- In lines 322-325: “Therefore, a pH of 6.8 was selected because, at this pH, 81.7% of vanillin is in the protonated form and the ferulic acid and vanillic acid are still, mostly, in their non-protonated form (>99%) (Table 6), which makes vanillin recovery possible free from non-volatile ferulic acid and vanillic acid.”
In Figure 3, with the pervaporation-fractionated condensation system, no contaminants were found in the permeate. Also, in Figure 4, with the vacuum distillation-fractionated condensation system, no contaminants were found in the permeate (with the exception of traces of vanillyl alcohol, which does not represent a problem since it is also present in botanical vanillin).
The other parameters referred by the reviewer are addressed in the answers to comments 2) and 3), presented below:
- Vacuum pressure affects the boiling point of the constituents in the bioconversion broth while the pH of the bioconversion broth affects the volatility of the constituents in the bioconversion broth. Vacuum pressure dominates the vanillin permeate flux while pH controls the containments of the vanillin permeate. The author needs to address more in adjusting the pH and vacuum pressure despite considering the energy consumption of the vacuum pump.
To address the value of vacuum pressure in pervaporation and the energy consumption of the vacuum pump, the following texts were added:
- In lines 399-407: “The values of permeate pressure and temperature of the first condenser was previously optimised [18]. The value of vacuum pressure selected for the pervaporation-fractionated condensation system was 1.5 mbar., considering the best balance between the required driving force for the permeation of vanillin and the energy input associated, and the average value of permeate pressure for all pervaporation experiments performed The defined permeate pressure and temperature of the first condenser was were, respectively, 1.5 mbar of permeate pressure (average value of permeate pressure for all pervaporation experiments performed) and at 0 ºC, considered sufficient to assure a complete condensation of vanillin.”
- In lines 607-611: “Also in pervaporation, the energy consumption of the vacuum pump is rather small, and may be considered negligible, when compared with the energy for evaporation/pervaporation. Indeed, after setting the vacuum pump at 1.5 mbar, this pressure remains almost constant without pump work because the water and vanillin fluxes are rather mild and non-condensable gases are not present and, therefore, there is no need for pump work to remove them.”
The energy consumption calculated in this work was “the thermodynamic energy necessary to produce a Kg of vanillin” (in lines 599-601), which is much higher than the energy consumption of the vacuum pump.
- The author needs to explain why the addition of consecutive water pulses in the feed can promote more vanillin permeate yield.
The effect of the addition of consecutive water pulses in the feed was explained in the original manuscript:
- In lines 490-496: “After evaporation of water, vanillin becomes more concentrated (and consequently with a lower activity coefficient in the concentrated aqueous solution), which lowers its volatility in solution. Therefore, in order to increase the rate of evaporation of vanillin without increasing the temperature, a strategy of multiple water feeding pulses was implemented. This approach consists of adding several consecutive water pulses, each one with the same mass of the evaporated water, keeping constant the total mass of the feed solution.”
Although the addition of water in the feed may promote vanillin yield, the pH in the bioconversion broth may change.
The addition of water did not change the pH value of the bioconversion broth. The variation of pH was always below 0.1 units, which is a negligible variation.
The author needs to prove no containments in the vanillin permeate after the water is added to the feed.
The absence of contaminants was confirmed by GC-FID and NMR analyses.
- The normal practice in VOC separation from a mixture stream in some consecutive condensers is to condense water at the first condenser. In this study, only 46% vanillin was recovered in the first condenser. To separate vanillin from water The temperature setting of the first condenser is to separate volatile constituents of the bioconversion broth from water, the consecutive condenser then is used to separate vanillin from potential containments.
This work demonstrates a proof-of-concept for the recovery and purification of vanillin. Since the processes were developed at laboratory scale, with an over-dimensioned vacuum pump, the residence time of the permeate stream in the condensers was significantly lower than usual in industrial processes, which explains the low % of vanillin recovered in the first condenser. This is why we wrote:
- In lines 634-640: “Future work at pilot scale requires the design of condensers with optimised area/residence time of condensable compounds, in order to assure a high recovery of the permeated vanillin. These pilot studies should be performed for both approaches discussed in this work. Considering the actual energy cost of 0.10 € KWh-1 for industrial production and the current risk of increasing energy price, the pervaporation-fractionated condensation approach is recommended. It should be also emphasised that this approach does not involve the use of solvents with inherent costs of solvent reuse/disposal.”
5) In this study, only the energy of vacuum pumps was compared. The author agrees that the energy consumption of the condensers is significant in vacuum vanillin recovery systems. To exclude the energy consumption of condensers from the investigated system is incomplete in the assessment of the system performance.
The reviewer is right and we are aware that the value calculated is not a full estimation of the energy input required. Still, it should be mentioned that the energy for evaporation/pervaporation represents (under industrial operation) the largely dominant fraction of the energy required. This issue was addressed in answer to comment 2). As clearly stated in the manuscript, the calculated energy consumption was the “the thermodynamic energy necessary to produce a Kg of vanillin” (in line 599-601).
To better addressed this issue, see the 2nd text in comment 3).
